# Changes in Wavefront Error of the Eye for Different Accommodation Targets under the Application of Phenylephrine Hydrochloride

María Mechó-García [1,2,*], Iñaki Blanco-Martínez [1,2], Paulo Fernandes [1,2], Rute J. Macedo-de-Araújo [1,2], Miguel Faria-Ribeiro [1,2] and José Manuel González-Méijome [1,2]

1 Clinical & Experimental Optometry Research Laboratory, Centre of Physics (Optometry), School of Sciences, University of Minho, 4710-057 Braga, Portugal
2 Physics Center of Minho and Porto Universities (CF-UM-UP), 4710-057 Braga, Portugal
* Correspondence: mmechogarcia@fisica.uminho.pt

**Abstract:** Pharmacological dilation of the eye to have a larger pupil diameter may allow a better understanding of the wavefront error changes with accommodation. This work aimed to investigate whether dilation of the pupil with Phenylephrine Hydrochloride (PHCl) application changes the accommodative response and the Zernike coefficient magnitude with accommodative demand when computed to a common pupil size. Sixteen right eyes of healthy young subjects were measured with the commercial Hartmann–Shack aberrometer IRX3 (Imagine Eyes, Orsay, France) 30 min after two drops of 1.0% PHCl were applied. The eye wavefronts for accommodative demands from 0 to 5 D were measured in natural conditions and after pupil dilatation. Statistically significant differences between both conditions were found for the Zernike coefficients $C_3^{-1}$, $C_3^1$, $C_4^0$ and $C_6^0$. Without the effect of PHCl, higher values were found for all higher-order Zernike coefficients (HOA). With increased accommodative response, an increase of $C_3^{-1}$ and a decrease of $C_3^1$ was observed and the $C_4^0$ becomes more negative; the change from positive to negative is shown in the accommodative demand of 1.5 D. Conversely, $C_6^0$ increases with increasing accommodative demand. To conclude, the results demonstrated that the mydriatic effect of PHCl causes changes in the magnitude of HOA when accommodation is stimulated. The trends observed in the different Zernike coefficients were the same reported in previous studies.

**Keywords:** accommodation; mydriatics; phenylephrine; wavefront aberrations; Zernike coefficients

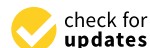



## 1. Introduction

In clinical practice, aberrometry is used to objectively measure the optical quality of the eye [1]. Therefore, ocular aberrations have been measured in a large population when accommodation is stimulated with different methods and devices [2–4]. When ocular accommodation is induced, changes in ocular aberrations are expected to occur due to changes in the eye's internal structures.

During accommodation, the ciliary body is seen to move forward, and the zonular fibres lengthen. These fibres, which are connected to the vitreous body and the lens posterior capsule, allow the lens shape to change. The curvature and thickness of the lens are altered due to the contraction of the ciliary body, which decreases the radius of the anterior and posterior lens. The changes in the posterior surface are smaller [5], the lens becomes thicker and the depth of the anterior chamber decreases [6]. Pupil diameter decreases as the eye accommodates and converges as a part of the near triad [7]. As a consequence, as the eye accommodates and the pupil aperture decreases, the wavefront area available to be measured is reduced, so the optical quality information obtained with a wavefront aberrometer is limited.

Increasing pupillary size (mydriasis), without affecting the accommodative response of the eye, may be of interest in various clinical applications. Phenylephrine hydrochloride (PHCl) is a mydriatic that acts directly on the α-receptor of the pupillary dilator muscle and is commonly used for a mydriatic effect [8]. Generally, this drug has its maximum effect 30 min after installation of the first drop. Additionally, no effect on the subject's accommodative state is expected. There is, however, some controversy with studies reporting a reduction in the accommodative response after the instillation of PHCl, while others have reported no change in the accommodative response. In 1956, Biggs and colleagues [9], using a subjective Badal optometer, were the first to describe a mean reduction of 0.66 D in the accommodation amplitude after the instillation of 8 drops of PHCl with a concentration of 10%. Later, in 1986, Mordi and colleagues measured the accommodative response with an infrared optometer, instilling two drops of PHCl with a concentration of 2.5%, and found a decrease in accommodative response of approximately 50% two hours after the instillation of the first drop [10]. Two years later, Zetterström [11] reported significant differences in the ability to accommodate between three different concentrations of phenylephrine (0.1%, 1.0% and 10%). The accommodative response was measured using Donder's push-up test and using an artificial pupil of 2 mm. They concluded that PHCl influences an average of 2.00 to 3.00 D in the accommodative response. Later on, Gimpel and colleagues [12] observed a decrease of 1.22 D and 1.39 D, 30 min and 60 min after the instillation of PHCl 2.5, respectively. In 2002, with the same PHCl concentration, Do and colleagues [13] reported a decrease of 17% in the accommodative amplitude measured by a subjective method, while no significant differences were observed when measured by an objective method.

Despite the effects of PHCl on accommodative response previously reported, other studies did not describe any changes. Leibowitz and Owens [14] reported no systematic changes in the accommodative response measured with a laser optometer with a Badal system, 25 min after the instillation of PHCl 10%. A few years later, Eyeson-Annan et al. [15] performed a study to identify any significant difference between accommodation and maximum pupil dilatation. Three drops of PHCl 10% were instilled, and no significant effect on the amplitude of accommodation was obtained 20 min after instillation. Later, Ostrin and Glasser [16] reported no significant effects of PHCl on accommodative amplitude, dynamics or resting position when the accommodative responses under the 10% PHCl concentration effect were evaluated using electrical stimulation of the Edinger–Westphal nucleus in rhesus monkeys. In 2012, Richdale and colleagues [8] measured the objective and subjective accommodative response using one drop of 2.5% PHCl, and reported a reduction of 1.0 D in the accommodative response only when measured by subjective methods. More recently, Del Águila Carrasco et al. [17] measured static (using a commercial aberrometer) and dynamic (with a sinusoidal moving stimulus at 10 Hz) accommodation after the instillation of 10% PHCl. They reported that the difference in the accommodative ability after the instillation of PHCl depends on the method used to calculate the static and dynamic accommodative response. The calculated minimum RMS refraction was significantly different before and after PHCl instillation for both static and dynamic accommodation. However, for paraxial refraction, no significant differences were found between both.

As described above, studies report a decrease in the accommodative response after the instillations of PHCl, while others suggest that there is no effect. These contradictory conclusions cause controversy regarding the effect produced after PHCl instillation in the accommodative response. The differences in the methodology used, the concentration of PHCl and number of drops installed, as well as sample characteristics may explain these contradictory results. Despite this previous evidence on the potential effect of phenylephrine on accommodation, the present work shows a further detailed evaluation of the wavefront information rather that an evaluation of the refractive error and other less detailed metrics of refraction. This study was considered to be able to provide detailed information on the changes in wavefront error induced by the instillation of phenylephrine to dilate the pupil and clinicians and scientists should be aware that, although it is consid-

ered that this substance does not affect accommodation significantly, our results show that when evaluating wavefront error, these might become significant.

The main goal of the present study is to investigate the changes in the crystalline lens with accommodation by indirect observation of the changes in the wavefront aberrations under the effect of PHCl. As the pupil diameter decreases as the eye accommodates, in order to better understand this process, it is useful to observe these changes with a larger pupil. Therefore, to obtain a larger pupil, a mydriatic agent (PHCl 1.0%) will be instilled to see if it alters the subjects' accommodative response. A secondary goal is to analyze the behavior of the different Zernike coefficients that were analyzed for each accommodative demand.

## 2. Materials and Methods

In the current study, we analyzed the differences in the accommodative response calculated from wavefront data, before and after PHCl administration.

Sixteen young adults aged 20 to 30 years were recruited. Inclusion criteria included spherical ametropia less than 8.00 D, astigmatism less than 1.00 D and best corrected visual acuity (BCVA) equal to 1.0 logMAR or better. None of the subjects had any ocular pathology or had undergone any refractive surgery. Measurements were taken only in the right eye, due to the presence of optical enantiomorphism (mirror symmetry) and the high correlation between both eyes. This research followed the tenets of the Declaration of Helsinki and was approved by the Ethics Subcommittee for Life and Health Sciences of the University of Minho. Informed consent was obtained from subjects after an explanation of the possible consequences of the study.

Primary outcome measurements were BCVA with the EDTRS chart, and monocular amplitude of accommodation (AA) with the Donders' push-up method, which consists of bringing a target closer and closer to the patient's eye until it first blurs. Intraocular pressure (IOP) was measured with the Ocular Response Analyzer (ORA) as a control, due to the possible risk of the angle closure after instilling PHCl.

The ocular wavefront aberrations were assessed using the Irx3 wavefront aberrometer (Imagine Eyes, Orsay, France), which is based on the Hartmann–Shack technique [18]. This instrument has a $32 \times 32$ lenslet array and uses 780 nm wavelength to illuminate the eye. The built-in fixation target designed for central measurements consists of a black 6/12 Snellen letter "E" on a white background, subtending a total field of about $0.7 \times 10$ degrees, with luminance of 85 cd/m$^2$.

To measure ocular aberrations, first, the least accommodated state of the eye was measured with the automatic iterative fogging procedure [19]. The aberrometer's internal Badal system permits one to estimate the equivalent sphere of the subject. The equivalent sphere measurement was saved and placed as the initial target vergence. The stimulus target vergence was then increased from 0.0 D (target placed at the remote point of the subject) to 5.0 D by steps of 0.5 D (11 steps). The measured eyes fixated on the target, while the contralateral eye was occluded, to ensure a larger pupil in the measured eye. The subject was asked to keep looking at the smaller visible detail in the stimulus target. For each step, three repeated measures were obtained, making thirty-three measurements in total for each subject. After each step, the target movement paused for 2 s to allow enough time for the subject's accommodation to respond. After completing the experiment under natural pupil conditions, the same right eye was dilated using a topical mydriatic drug (Davinefrina, DAVI, Barcarena, Portugal). Two drops of a 1.0% phenylephrine hydrochloride solution were instilled, separated by a 5-min interval. A total of 30 min after the first instillation, the same sequence of measurements was repeated under artificial dilatation. All the measurements were performed under dim room illumination. During the stepwise accommodation stimulation, which took in total 22 s, the subject was allowed to blink to avoid an increase in aberrations introduced by air–tear film interface changes [20,21].

All the analyses were reported according to the Optical Society of America (OSA) [22,23] recommended standards. The resulting wavefront response data were expressed in the form

of a Zernike expansion up to 6th order, rescaled to a 4 mm pupil size, corresponding to the smaller diameter measured in natural conditions.

Normality of the data was checked using Shapiro–Wilk test. To compare differences in the accommodative response before and after phenylephrine administration, paired *t*-tests or Wilcoxon tests were used, if the sample was parametric or non-parametric, respectively. The effect size was calculated with Hedges' correction. Subsequently, to identify differences between the measurements (i.e., fixed bias) or possible outliers, a box plot of the difference of the measurements with and without phenylephrine for the different Zernike coefficients was represented.

## 3. Results

Sixteen subjects were included with a spherical refractive error ranging from $+0.50$ D to $-6.00$ D, with a mean $\pm$ SD of $-2.46 \pm 2.06$ D (without PHCl) and a mean of $-2.50 \pm 2.07$ D (with PHCl). The mean $\pm$ SD pupil diameter was $6.61 \pm 0.98$ mm and $7.42 \pm 0.54$ mm before and after PHCl instillation, respectively.

The average of the subjectively assessed amplitude of accommodation was $10.84 \pm 0.79$ D (ranging from 10 D to 12 D).

### 3.1. Pupil Size

Figure 1 shows the mean value of the pupil diameter obtained for each subject (before and after PHCl instillation). An increase in pupil size was observed after the instillation of PHCl. It should be noted that these are the pupil diameters for 0 D accommodative state. The differences between both conditions will increase with accommodative demand. On average, the pupil size under the effects of PHCl was 0.811 mm larger than the average pupil size under natural conditions. Although measurements were performed under the same lighting conditions and the effect of the mydriatic was controlled to be the same for all (in terms of time), it can be observed that there are differences between natural conditions and after the instillation of PHCl, which reached statistically significant differences ($p$-value $< 0.05$) in some subjects.

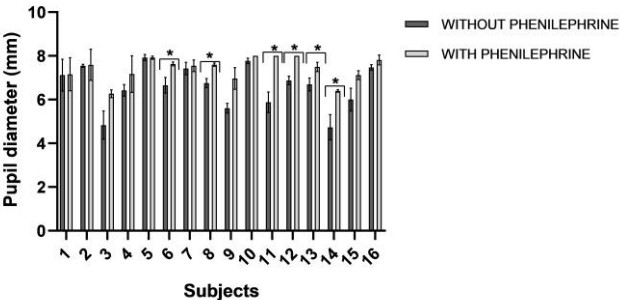

**Figure 1.** Representation of the pupil diameter in the different subjects measured, with and without the effect of PHCl. Error bars represent the standard deviation (SD). * Simple paired *t*-tests, statistically significant ($p$-value $< 0.05$).

### 3.2. Zernike Coefficients

Table 1 shows the $p$-values obtained for each accommodative demand between both conditions.

Figure 2 shows the mean of the defocus $C_2^0$ Zernike coefficient obtained for each accommodative demand in both conditions studied. There is a positive increase in $C_2^0$ with accommodative demand. No difference between both examination conditions can be observed. Table 1 shows the $p$-values obtained for each accommodative demand for both conditions.

**Table 1.** Wilcoxon test or paired *t*-test values obtained for Zernike coefficients $C_2^0$, $C_3^{-1}$, $C_3^1$, $C_4^0$ and $C_6^0$ for each accommodative demand between measurements carried out with and without PHCl instillation conditions.

| Accommodative Demand (D) | *p*-Value C (2, 0) | *p*-Value C (3, −1) | *p*-Value C (3, 1) | *p*-Value C (4, 0) | *p*-Value C (6, 0) |
|---|---|---|---|---|---|
| 0 | 1.0 | 0.13 | 0.88 | 0.22 | 0.09 |
| 0.5 | 0.35 | 0.33 | 0.41 | 0.45 | **0.04 *** |
| 1.0 | 0.09 | 0.24 | 0.50 | 0.38 | 0.06 |
| 1.5 | 0.28 | 0.08 | 0.96 | 0.24 | **0.01 *** |
| 2.0 | 0.15 | 0.09 | 0.35 | 0.14 | **0.02 *** |
| 2.5 | 0.41 | 0.05 | 0.35 | 0.09 | 0.08 |
| 3.0 | 0.25 | 0.07 | 0.11 | **0.02 *** | **0.01 *** |
| 3.5 | 0.36 | **0.01 *** | 0.16 | **<0.001 *** | **<0.001 *** |
| 4.0 | 0.55 | **0.02 *** | **0.02§** | **0.02 *** | **0.02 *** |
| 4.5 | 0.77 | **0.02 *** | **0.01§** | **0.02 *** | **0.01 *** |
| 5.0 | 0.83 | **0.01 *** | **0.02§** | **0.02 *** | **0.01 *** |

§ Wilcoxon test, statistically significant (*p*-value < 0.05). * Simple paired *t*-tests, statistically significant (*p*-value < 0.05). Statistically significant differences are presented in bold type.

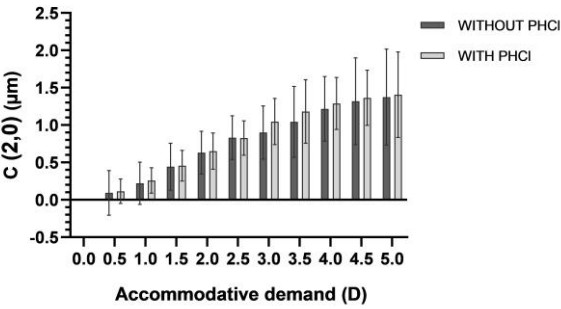

**Figure 2.** Defocus Zernike coefficients for the different target vergences, with and without phenylephrine (grey and black bars, respectively), calculated for the same 4 mm pupil diameter. Error bars represent the standard deviation (SD).

Under natural conditions, vertical coma $C_3^{-1}$ showed a positive increase with accommodative demand (Figure 3a, black bars). This increase was smaller after the instillation of PHCl (Figure 3a, grey bars). The differences were statistically significant for accommodative demands above 3.5 D (Table 1). The effect size was g < 0.06, which is correspondent to a small effect size.

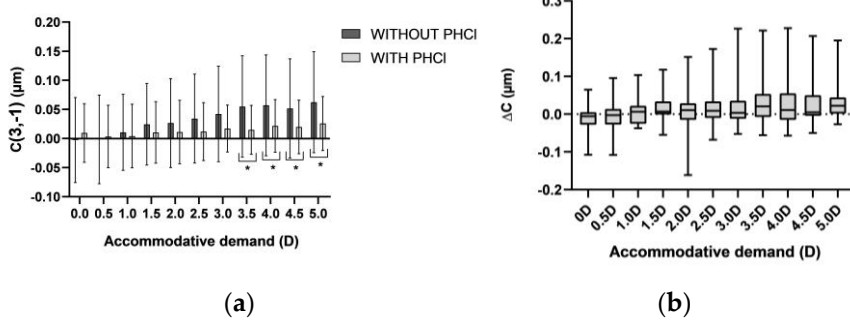

(**a**)                          (**b**)

**Figure 3.** (**a**) Zernike vertical coma coefficients for the different accommodative demands, before and after the instillation of PHCl (grey and black bars, respectively). (**b**) Box plot of the difference in Zernike coefficients between natural conditions and after instillation of PHCl for each accommodative demand. * Denote statistically significant values (*p* < 0.05). Error bars represent the standard deviation (SD).

Moreover, horizontal coma $C_3^1$ had a negative increase with accommodative demand (Figure 4a, black bars). Under the effect of PHCl (Figure 4a, grey bars), an increment in the positive values was observed when the accommodative demand increased. Statistically significant differences between both conditions were observed for accommodative demands (*p*-value < 0.02), in accommodative demands from 4.0 to 5.0 D.

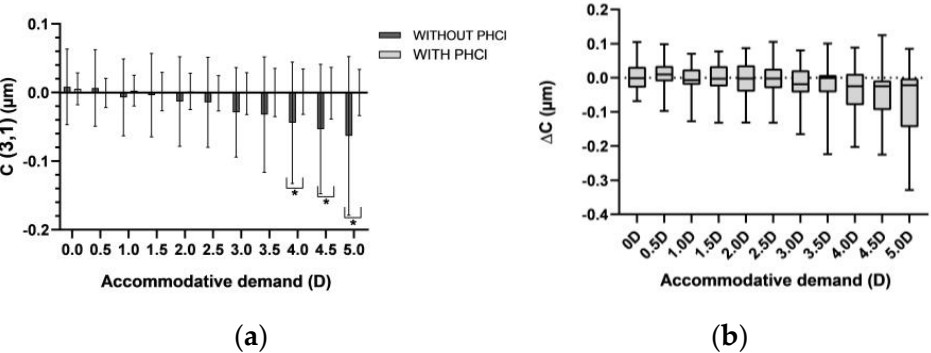

**(a)**          **(b)**

**Figure 4.** (**a**) Horizontal coma Zernike coefficients for the different target accommodations, with and without PHCl (grey and black bars, respectively). (**b**) Box plot of the difference in Zernike coefficients between natural conditions and after instillation of PHCl for each accommodative demand. * Denote statistically significant values (*p* < 0.05). Error bars represent the standard deviation (SD).

Measurements of fourth-order spherical aberration $C_4^0$ taken in natural conditions showed a shift to more negative values with increased accommodative demand (Figure 5a, black bars). Measurements taken after the instillation of PHCl also showed a negative shift, but to a lesser extent (Figure 5a, grey bars). These differences between both conditions were statistically significant for accommodative demands above 3.0 D (*p*-value < 0.02).

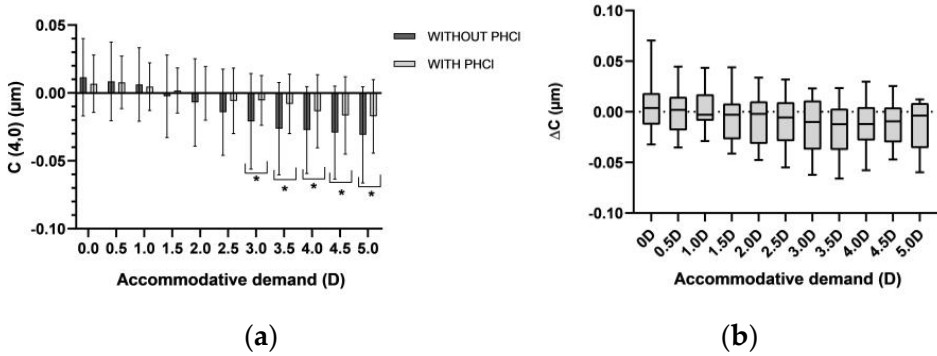

**(a)**          **(b)**

**Figure 5.** (**a**) Primary spherical aberration Zernike coefficients for the different target accommodations, with and without PHCl (grey and black bars, respectively). (**b**) Box plot of the difference in Zernike coefficients between natural conditions and after instillation of PHCl for each accommodative demand. * Denote statistically significant values (*p* < 0.05). Error bars represent the standard deviation (SD).

Sixth-order spherical aberration $C_6^0$ showed an increase with accommodative demand, with larger values when measurements were taken under natural conditions (Figure 6). Differences were statistically significant for most accommodative demands (*p*-value < 0.04), except for 0, 1.0 and 2.5 D. The effect size was small, with g < 0.05.

The accommodative response among the sixteen subjects is slightly greater under the pupil in natural conditions. There was a difference in accommodative response between the subjects studied of 1.56 D under natural conditions and 2.10 D under the effect of PHCl. Thus, there was a greater variability of the accommodative response when the subjects are under the effect of PHCl (Figure 7).

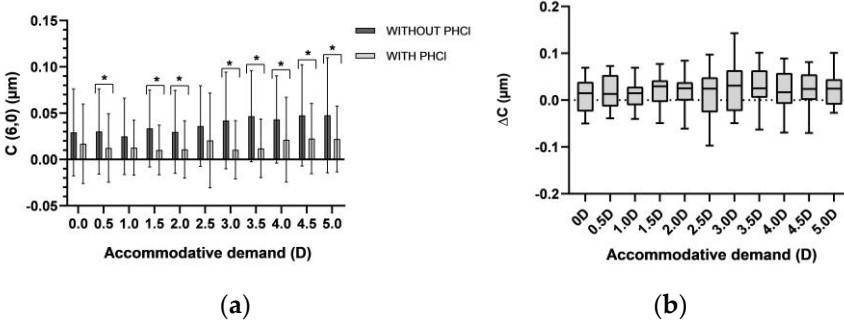

(**a**)                                         (**b**)

**Figure 6.** (**a**) Secondary spherical aberration Zernike coefficients for the different target accommodations, with and without PHCl (grey and black, respectively). (**b**) Box plot of the difference in Zernike coefficients between natural conditions and after instillation of PHCl for each accommodative demand. * Denote statistically significant values ($p < 0.05$). Error bars represent the standard deviation (SD).

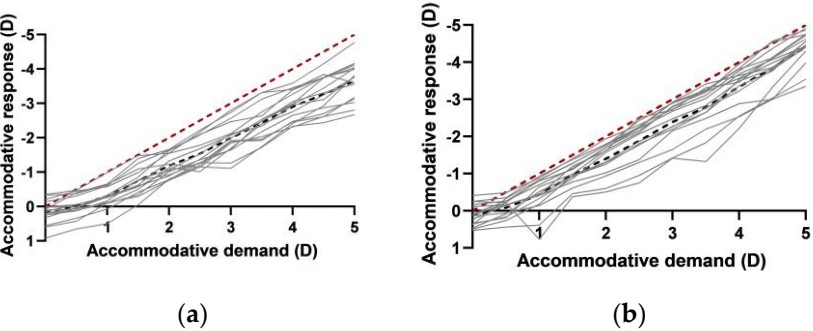

(**a**)                                         (**b**)

**Figure 7.** Accommodative response in terms of sphere plotted for each subject. (**a**) Pupil under natural conditions and (**b**) pupil under the effect of PHCl, for each subject. Both of them plot for the same pupil size, 4 mm. The dashed red line expresses the expected values of accommodative response. Additionally, the dashed black line expresses the mean values for each accommodative demand.

## 4. Discussion

Understanding the wavefront profile when accommodation is active has potential applications in the area of optometry and ophthalmology (contact lenses, possible surgical restoration of the accommodation and refractive surgery, among others). So, knowing the wavefront pattern at maximum pupil size may help us to understand wavefront error changes in the periphery of the crystalline lens.

There are different technologies to measure the wavefront of the eye. These technologies can be divided into different groups according to the way data are collected. Subjectively or objectively, with single-pass systems or double-pass systems or based on the serial (Ray Tracing or Automatic Retinoscope) or parallel (Hartmann–Shack or Tschering) principle, referring to a 1-by-1 measurement of the data points or simultaneous collective measurement of all points, respectively. As the parallel methods are faster in obtaining the wavefront, this was chosen to measure the ocular aberrations that occur in the eye when the subject's accommodation was stimulated [23].

There are some technical difficulties when measuring aberrometry with a Hartmann–Shack sensor, as it has a strong dependence on the position of the image points, so if the eye is very aberrated, the points overlap and the software cannot distinguish where the point has originated and causes a reduction in the accuracy of the aberrometry [24]. In addition, corneal irregularities and a poor tear film can lead to an increase in aberrations, especially higher-order aberrations [20,21].

This study had two main objectives. The first one was to verify the possible variation in the magnitude of the Zernike coefficients when the accommodation was stimulated up to 5.0 D, after instilling two drops of PHCl with a concentration of 1.0%. The second one

was to describe the behavior of the different Zernike coefficients that were analyzed for each accommodative demand.

The first 25 Zernike coefficients (up to the sixth order) were studied, but only the Zernike coefficients that were found to be statistically significantly different and present a significant change when accommodation was stimulated were described, as suggested by previous studies [19,25–27].

The results showed statistically significant differences in all the high-order Zernike coefficients that were analyzed. The horizontal coma had little effect; the values were near to 0 μm under the effect of PHCl.

Assuming that PHCI does not affect accommodation, a higher accommodation response with a larger pupil size is expected, as the depth of focus decreases for larger pupil sizes. This is not in accordance with the results obtained. In fact, most of the studies reported the same results. The pupil dilates after PHCl instillation and the reduction in the depth of focus causes blur that is translated as a reduction in the accommodative response. However, the optical factors associated with mydriasis did not respond to this blur, and other effects may be the cause of this behavior, including the direct action of PHCl in the ciliary body. As the accommodation is mostly controlled by the action of the ciliary muscle, and PHCl is a primarily $\alpha$-adrenoceptors agonist, this should have an influence on the $\beta$-adrenoceptors located in the ciliary body by altering the size and function of ciliary muscle via these proteins. Likewise, the differences observed in the Zernike coefficients between both conditions may also be due to a change in the location of the pupil center with pupillary dilation, its position being closer to the geometric center of the cornea at maximum dilation [28–30].

Studies have reported different conclusions when evaluating the effect of phenylephrine on accommodation. However, they are not comparable with the results obtained in the present study because of the use of different concentrations of phenylephrine [11,30–32], different times of drug effect [33,34] and different methods of measuring the accommodative response used [9,31]. Looking at the results obtained in the present study, it can be concluded that the magnitude in the high-order Zernike coefficients was altered after instilling PHCl at 1.0% concentration.

The second objective was to observe the trends in the different Zernike coefficients studied for the eleven accommodative demands. All Zernike coefficients were rescaled to a common 4 mm pupil size, and the measurements were taken under natural conditions.

An increase in the vertical coma $C_3^{-1}$ was found when the accommodative demand increased (Figure 3a), starting at negative values up to the accommodative demand of 1.0 D and shifting to less negative values up to −0.031 μm for the maximum accommodative demand, and up to 0.026 μm after the PHCl effect. The Zernike coefficients are practically constant from the accommodation demand of 3.0 D in both conditions examined, this may be due to the fact that upon accommodation, the zonule supporting the lens relaxes, causing a slightly lower decentration of the lens [35].

In contrast, horizontal coma $C_3^1$ showed a decrease when the accommodative demand increased (Figure 4a), starting at the positive values up to the accommodative demand of 1.0 D, and shifting to negative values up to −0.062 μm at 5.0 D of accommodative demand, and up to −0.001 μm under the PHCl effect. $C_3^1$ turning to negative values may imply an increase in nasal-temporal asymmetry in the lens geometry due to the accommodation–convergence relationship. These changes in the Zernike coefficients were not in agreement with those found in the literature, which reported changes in both coma coefficients with the accommodation, but they do not systematically change [19] and remain relatively stable during different accommodative demands [36].

The most described change in Zernike coefficients with accommodation is in primary spherical aberration ($C_4^0$). As in agreement with many studies, (C40) described a general trend, which becomes more negative as the accommodative demand increases (Figure 5a), $\Delta C_4^0 = -0.041$ μm from 0 D to 5 D under natural pupil conditions. Similar results were obtained by López-Gil et al. [19] (2008), $\Delta C_4^0 = -0.044$ μm from 0 D to 5.0 D,

Radhakrishnam and Charman [37] (2007), $\Delta C_4^0 = -0.048$ μm from 0 D to 2.5 D, and Plainis et al. [38] (2005), $\Delta C_4^0 = -0.054$ μm. A minor decrease under the PHCl effect was obtained, $\Delta C_4^0 = -0.023$ μm. The C (4,0) changes from positive to negative values at an accommodative response of 1.5 D, which is comparable with the findings of 1.0 to 1.5 D by Jenkins (1963) [39], 2.0 D by Atchison et al. (1995) [40], and 1.7 D by H. Cheng et al. (2004) [41]. This may be due to the change in the geometry of the lens surfaces; as the central thickness of the lens increases, the conicity changes, inducing negative spherical aberration [42,43]. This behavior of $C_4^0$ has been described in in vivo measurements during accommodation in rhesus monkey eyes [16], in isolated crystalline lenses [44] and in measurements made on subjects using different wavefront measurement devices [26,27,36,37,44,45].

Secondary spherical aberration (Figure 6a) showed an increase of 0.017 μm from 0 D to 5 D when the pupil was in natural conditions, and an increase of 0.004 μm under the PHCl effect. These results were reported in other studies, always turning to the positives when the eye was accommodated [27] and being of different magnitudes depending on the subject [19,30,31].

Large pupils are necessary, appropriate and give more information about wavefront measurements. However, under normal conditions the pupil size will not be altered, so accommodative state and wavefront aberrations for the subjects measured may be more reliable. As pupil dilatation can become necessary in some applications, such as in the study of the ocular fundus particularly in myopic eyes, clinicians should be aware that the wavefront and the accommodative response can change using pharmacological pupil dilation with phenylephrine, even when the accommodation is not intentionally paralyzed by the cycloplegic agent.

If the results obtained in the present study are confirmed in future studies, they should be taken into account when performing wavefront-guided refractive surgery, as mydriatic agents are used in these surgeries to allow for a larger working area. Moreover, when detecting pathologies that affect the magnitude of the Zernike coefficients, these should not be masked by the effect of the mydriatic agent, as the tendency patterns are the same as with the pupil under normal conditions. In addition to this, analyzing the magnitude of the Zernike coefficients will allow the detection of early pathologies such as corneal ectasia, and abnormalities in the accommodative response when studying the magnitude of the Zernike coefficients with accommodation.

## 5. Conclusions

To conclude, it can be observed that the accommodative response was better in subjects under natural pupil conditions (Figure 7). As previously reported [9–13], these results confirm that there is a change in the magnitude of the Zernike coefficients after PHCl mydriatic drug instillation when the accommodation is stimulated. The trends observed in the different Zernike coefficients, for a fixed pupil diameter of 4 mm, were the same reported before [19,36–45] for $C_2^0$, $C_3^{-1}$, $C_4^0$ and $C_6^0$. The horizontal coma $C_3^1$ was not in agreement with the literature results, so further studies must be carried out to ensure that the effect of this trend is linked with the accommodation effect.

Thus, characterization of the changes that occur in ocular aberrations when accommodation is stimulated will provide interesting information for customized wavefront corrections when refractive surgery is performed, for instance, in wavefront-guided refractive surgery with the aim to correct both sphero-cylindrical refraction and HOA. In addition, the use of aberrometry to assess the accommodative response of subjects is of great interest, as it allows one to study the interaction between the different Zernike coefficients as the subject accommodates, and to observe possible abnormalities, such as accommodative lag or lead.

**Author Contributions:** Conceptualization, J.M.G.-M. and M.M.-G.; methodology, J.M.G.-M. and M.M.-G.; formal analysis, M.M.-G. and I.B.-M.; investigation, M.M.-G.; resources, M.M.-G., I.B.-M. and J.M.G.-M.; data curation, M.M.-G. and I.B.-M.; writing—original draft preparation, M.M.-G.; writing—review and editing, M.M.-G., I.B.-M., P.F., M.F.-R., R.J.M.-d.-A. and J.M.G.-M.; visualization, M.M.-G.; supervision, P.F., M.F.-R., R.J.M.-d.-A. and J.M.G.-M.; project administration, J.M.G.-M.; funding acquisition, J.M.G.-M. All authors have read and agreed to the published version of the manuscript.

**Funding:** This project has received funding from the European Union's Horizon 2020 research and innovation programme under the Marie Skłodowska-Curie grant, agreement No. 956720.

**Institutional Review Board Statement:** The study was conducted in accordance with the Declaration of Helsinki and approved by the Ethics Committee for Research in Life and Health Sciences of the University of Minho CEICVS 081/2022, for studies involving humans.

**Informed Consent Statement:** Informed consent was obtained from all subjects involved in the study.

**Data Availability Statement:** The data used to support the findings of this study are available from the corresponding author upon request.

**Acknowledgments:** The European Union's Horizon 2020 research and innovation programme under the Marie Skłodowska-Curie grant.

**Conflicts of Interest:** The authors declare no conflict of interest.

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
