# Peer review of "Changes in Wavefront Error of the Eye for Different Accommodation Targets under the Application of Phenylephrine Hydrochloride"

_photonics, doi:10.3390/photonics10040381_

Round 1

Reviewer 1 Report

This is a well-written manuscript presenting a well-conducted study of
moderate interest.
By indirectly observing wavefront changes caused by PHCl and analysing
the behaviour of various Zernike coefficients for each accommodative
requirement, the authors looked into the changes in the crystalline lens
that occur with accommodation.
The methods are clear and replicable and the results presented match the
methods described.
The findings described by the authors and the conclusions correlate with
the results.
The paper does not raise any major concerns.
The main issue is the novelty of this study; the authors need to clarify
what new knowledge their work adds to the current body of knowledge.
Another important issue that requires further discussion is the clinical
application and how the findings of this study can change our clinical
practice.

Author Response

Dear reviewer, thank you for reading the text carefully and for your comments.

This is a very relevant question considering the previous studies. Indeed, this work reaffirms that the use of phenylephrine affects the accommodative response, showing significant differences in the Zernike coefficients with accommodation. The results are therefore in agreement with those already published by Biggs and colleagues (1995), Mordi and colleagues (1986), Zettertröm, Gimpel and colleagues (1994), Do and colleagues (2002), and others mentioned in the introduction section of the manuscript. However, the present work shows a further detailed evaluation of the wavefront information rather that an evaluation of the refractive error and other less detailed metrics of refraction.

So, the present study provides detailed information on the changes in wavefront error induced by the instilation of phenylephrine to dilate the pupil and clinicians and scientists should be aware that although it is considered that this substance does not affect accommodation significantly, our results show that when evaluating the wavefront error, those might become significant. This has been incorporated in the Introduction section justifying the realization of this study.

Therefore, these alterations should be taken into account when instilling phenylephrine in clinical practice, being of particular relevance in pre-operative measurements to predict intraocular lens power.

Furthermore, using aberrometry to assess the accommodative response of subjects is of great interest, as it allows us to study the interaction between coefficients as the subject accommodates, and to observe possible anomalies, such as accommodative lag or lead. 

In addition to the assessment of accommodative response, aberrometry has quite powerful applications in clinical practice, such as wavefront-guided refractive surgery that aim to correct both the sphero-cylindrical refraction and higher-order aberrations (HOAs), the diagnosis of irregular astigmatism and, as a whole, to have an intuition of the optical quality of the eye. This part was added in the conclusions section of the manuscript.

Reviewer 2 Report

The authors studied whether or not the application of PHCl drops interferes with accommodation. For this purpose, they used aberrometry to study changes in the wavefront error of the eye. They concluded that after PHCl administration, accommodation is altered, confirming data already described in the literature. However, they note an alteration in horizontal ?31 coma, which is not in agreement with previous studies.

The work is well conducted, the statistical analysis is adequate.

Comments:

1) The authors should briefly describe the indications of aberrometry in ophthalmology beyond accommodation.
2) They should also briefly discuss the advantages of different aberrometers (e.g., wavefront sensor versus laser ray tracing, spatial resolution refractometer, etc.).
3) Can the results obtained if confirmed in future studies have a clinical implication, e.g., improve the outcome of refractive surgery?
4) Are there possible implications of the results obtained from the study in the diagnosis of eye diseases such as irregular astigmatism or others?
5) Finally, the possible technical difficulties of aberrometry should be commented on.

Author Response

Point 1: The authors should briefly describe the indications of aberrometry in ophthalmology beyond accommodation.

Response 1: Dear reviewer, thanks for your comments. The characterization of the changes in ocular wavefront aberrations with accommodation will also provide useful information to customize aberration correction using techniques such as refractive surgery and contact lenses. You can find this information added to the manuscript in the conclusion section.

Point 2: Discuss briefly the advantages of different aberrometers (e.g., wavefront sensor versus laser ray tracing, spatial resolution refractometer, etc.).

Response 2: Thank you for reading the text carefully. There are different technologies to obtain the aberrated wavefront of an eye. These different technologies can be divided in the way they collect the data, subjectively or objectively, with single-pass systems or double-pass systems, or divided based on the serial (Ray Tracing or Automatic Retinoscope) or parallel (Hartmann-Shack or Tschering) principle, referring, respectively, to a 1-by-1 measurement of the data points or simultaneous collective measurement of all points. As parallel methods can be fast, whereas serial methods require longer measurement time, it was decided to use this method to assess wave aberrations when accommodation was stimulated. This is discussed in the discussion section on page 11.

Point 3: Can the results obtained if confirmed in future studies have a clinical implication, e.g., improve the outcome of refractive surgery?

Response 3: This is an interesting question to consider. If the results obtained are confirmed in future studies, they should be taken into account when wavefront-guided refractive surgeries are performed under pupillary dilation. The development of a wavefront aberrometer makes it possible to measure  higher-order ocular aberrations. This information can be corrected in time via custom ablation or custom contact lenses. This answer can be found in the discussion section, on page 13 of the manuscript.

Besides this, the present results might be more relevant in young eyes with more active accommodation system. As pupil dilatation can become necessary in some applications such as in the study of the ocular fundus particularly in myopic eyes, clinicians should be aware that the wavefront and the accommodative response can change with pharmacological pupil dilation with phenylephrine, even when the accommodation is not intentionally paralyzed by a cycloplegic agent.   This has been added in the discussion section.

Point 4: Are there possible implications of the results obtained from the study in the diagnosis of eye diseases such as irregular astigmatism or others?

Response 4: Further studies will have to be done to have a clear conclusion to this question. The results obtained in the study may be relevant in the study of subclinical corneal ectasia as the magnitude of ocular aberrations in these pathologies is higher. However, irregular astigmatism can be detected in any of the other devices (by corneal topography, autorefractometer, etc).

On the other hand, the effect of phenylephrine should not mask any of the phenomena explained above, as the magnitude of the Zernike coefficients are altered, but the tendency is the same as when the measurements are made without phenylephrine. You can find this information added in the discussion section, page 13.

Point 5: The possible technical difficulties of aberrometry should be commented on.

Response 5: Thank you for such an interesting comment. The measurements of the wavefront aberrations have a strong dependence on the position of the spot image collection, so if the eye is highly aberrated, the spot will overlap and the computer can not distinguish where the individual point was originated, so this caused a reduction in the precision of the aberrometry measure. Moreover, corneal irregularities and poor tear film produce an increase in high-order aberrations. This discussion can be found on page 11 of the manuscript.

Besides this, clinical and research aberrometers are nowadays quite robust and reliable if appropriate care is taken with patient fixation alignment.
